# Epigenetic Regulation of Optic Nerve Development, Protection, and Repair

**DOI:** 10.3390/ijms23168927

**Published:** 2022-08-10

**Authors:** Ajay Ashok, Sarita Pooranawattanakul, Wai Lydia Tai, Kin-Sang Cho, Tor P. Utheim, Dean M. Cestari, Dong Feng Chen

**Affiliations:** 1Department of Ophthalmology, Schepens Eye Research Institute of Massachusetts Eye and Ear, Harvard Medical School, Boston, MA 02114, USA; 2Department of Medical Biochemistry, Oslo University Hospital, 0372 Oslo, Norway; 3Department of Ophthalmology, Oslo University Hospital, 0372 Oslo, Norway

**Keywords:** optic nerve, epigenetics, myelin, regeneration, oligodendrocytes

## Abstract

Epigenetic factors are known to influence tissue development, functionality, and their response to pathophysiology. This review will focus on different types of epigenetic regulators and their associated molecular apparatus that affect the optic nerve. A comprehensive understanding of epigenetic regulation in optic nerve development and homeostasis will help us unravel novel molecular pathways and pave the way to design blueprints for effective therapeutics to address optic nerve protection, repair, and regeneration.

## 1. Introduction

Waddington, in 1942, coined the term ‘epigenetics,’ which was defined as evident inheritable phenotypic alterations with no variations in genotype [1]. In the past two decades, the field of epigenetics has received immense attention owing to the discovery of several integral epigenetic factors that dictate gene transcription without any alterations to the DNA sequence but by adjusting the configuration of chromatin structure. The functionality of enzymes responsible for DNA and histone modifications eventually translates to altered transcriptional activity [2]. Epigenetics chiefly encompasses the events involving nuclear materials, namely chromatin accessibility, nucleosome positioning, histone modification, DNA methylation/demethylation, and enhancer-promoter interactions [2]. All these phenomena modulate the availability of DNA structure to the polymerase-mediated transcriptional activity resulting in gene expression or repression. In recent years, epigenetics has gained importance due to its proven role in development and pathophysiology [3,4]. Therefore, the current review aims at discussing the research around epigenetics in tissue protection, damage, and repair with a prime focus on the optic nerve—a vital tissue that connects the eye and brain to convey visual information and is essential for supporting visual functionality.

The optic nerve includes millions of nerve fibers originating from retinal ganglion cells (RGCs) that relay visual signals from the posterior eye segment to the brain. Incidents that attenuate the development or injury to the optic nerve and RGCs can lead to irreversible vision loss [5]. RGC axons channel from the retina to congregate and make up the optic nerve fibers. Embryonic RGCs are capable of undergoing axonal regeneration following injury [6]. However, this desirable characteristic is lost swiftly post-birth in almost all mammals. One possible explanation is the switch-off in the epigenetic program that controls optic nerve growth [7]. However, many attempts to restore the regenerative capabilities of the optic nerve in adults using epigenetic modification have been only moderately successful. Conditions affecting the optic nerve, including glaucoma [8], optic nerve atrophy [9], etc., have all been associated with epigenetic dysregulation. Moreover, the eye is considered a window for understanding complex brain functions and disorders [10,11], a phenomenon also linked to plausible epigenetic machinery and malfunction, which makes it even more important to gain knowledge related to these interlinked disorders.

Advancement in technology has led researchers to use novel state-of-the-art technologies to study epigenetic modifications during disease progression and in drug development. Some of these cutting-edge techniques that have been a useful tool in understanding optic nerve pathologies and related mechanisms include chromatin immunoprecipitation (ChIP) assay (DNA-protein interactions-chromatin structure analysis) [3], bisulfite sequencing (De novo DNA methylation exploration) [12]. Assay for Transposase-Accessible Chromatin (ATAC) with high-throughput sequencing (analyze chromatin accessibility across the genome) [13]. However, other advanced techniques such as Hi-C (epigenetic landscapes in 3D chromatin architecture analysis) and Nanopore sequencing have not yet been employed extensively in optic nerve research.

## 2. Epigenetic Modifications of DNAs and Histones

Epigenetic mechanisms are undoubtedly involved in neuronal differentiation, maturation, and synaptic network formation [14,15]. DNA methylation is pivotal in epigenetic modification that has the potential to tighten up the chromatin structure (Figure 1), thereby limiting the transcription ability of the cell. It involves the methyl group converting onto the C5 position of 5′-CpG-3′ dinucleotides, which results in the formation of 5-methylcytosine (5mC) [16]. This process is fueled by DNA methyltransferase (DNMTs), including DNMT1, DNMT2, DNMT3A, and DNMT3B, and the methyl group is actively contributed by S-adenosyl methionine (SAM). The complicated mechanism of DNA demethylation is mainly propelled by the ten-eleven translocation (TET) protein family. TET hydroxylases convert 5mC into 5-hydroxymethylcytosine (5hmC), thereby adjusting the methylation levels, which occur prior to cell division and multiplication [16]. This is vital because the newly formed cells are required to accumulate their own methylation marks for having their own specific characteristics and for optimal functionality. In humans, the three members of the TET protein family include TET1, TET2, and TET3, which are chiefly classified based on their structure and expression during development. Alternatively, histone can also undergo methylation and demethylation to regulate gene expression. Methylation takes place at different basic residues on histones and based on the magnitude of methylation and its location, it can lead to varying outcomes. The two main categories of histone methyltransferases are lysine-specific (SET (Su (var)3-9, Enhancer of Zeste, Trithorax) domain-containing or non-SET domain containing) and arginine-specific [17]. In both these types of histone methyltransferases, S-Adenosyl methionine (SAM) acts as the methyl donor group. Histone demethylases are categorized into two as well, namely amino oxidase homolog lysine demethylase (KDM) and JmjC domain-containing histone demethylases [18]. Histone methylation is in general associated with transcriptional repression, but methylation of certain lysine and arginine residues in histones leads to transcriptional activation.

These interdependent machineries of DNA and histone methyltransferases and demethylases work in synchrony to preserve the genomic methylation pattern (Figure 1). This critical balance of the methylation status significantly contributes to the induction and progression of various diseases, including vision loss due to optic nerve damage [8,19].

Histone acetylation is another epigenetic event that is unambiguously linked with amplified gene transcription [20]. During this process, histone acetyltransferases (HATs) will add a negatively charged acetyl group to lysine residues on histone proteins. The acetyl groups are removed by histone deacetylases (HDACs). Histone acetylation reduces the electrostatic affinity between histone proteins and DNA, which subsequently endorses a chromatin structure that is pro-gene transcription. HATs and HDACs are characterized based on their cellular localization and substrate preference. All these major epigenetic modifications regulate optic nerve development [21] and recent studies have unraveled their important role in optic nerve protection, regeneration, and repair [22,23,24,25].

## 3. Epigenetics in Optic Nerve and Retinal Development

During DNA replication, DNA methylation in the developing daughter DNA strand is maintained by DNMT1. DNMT3A and DNMT3B are responsible for “de novo” DNA methylation patterns in certain differentiating cell types [26,27]. Downregulating DNMT1 in cells has been shown to lead to a partial loss of DNA methylation [28], while DNMT3A and DNMT3B [29] are reported to co-express in the eye and have overlapping functions. DNMT2 is a multisubstrate tRNA methyltransferase and has been shown to support multi-tissue development, including the retina [30]. A retina-specific triple knockout mice model showed defective retinal development, validating the vitality of the three DNMTs in the eye [31].

DNA demethylation participates in the development and aging processes of the retina. A significant study aiming at safely reversing the senescence progression and restoring biological function using the eye as a model presented evidence that the ectopic expression of *Oct4* (also known as *Pou5f1*), *Sox2*, and *Klf4* genes (OSK), all of which are transcription factors, promote the re-establishment of the epigenetic scenario of aging neurons in the CNS [12]. The study validates that active demethylation and associated activity of transcriptional factors regulate the course of senescence and its functional reversal. Myelination is an established and comprehensively studied process. OPCs in neonates have several epigenomic regulators actively functioning for this process to occur, including histone deacetylation and repressive histone methyltransferase action [32]. On the contrary, in adults, OPCs retain the property of remyelination during injury or any biological insult. This beneficial property of remyelination becomes less efficient owing to aging. External factors, including extracellular matrix alteration and declining growth factor levels, have all been implicated in the age-dependent failure of the efficient remyelination process.

Myelin is the dedicated membrane sheet spread out from oligodendrocytes (OLs) that cover the optic nerve fibers. OLs, undergo differentiation from oligodendrocyte progenitor cells (OPCs), a process controlled through the interplay between transcription factors and epigenetic regulators [33,34]. This relationship can be modulated by various external stimuli that affect age and disease. OPCs differentiate at the end stage of CNS development and a major hallmark of their maturation is the accumulation of repressive histone K9 and K27 methylation marks [35]. Whilst the differentiation and proliferation of OPCs in neonates are regulated by DNMT 1, it only has minor effects during myelin repair in adult OPCs. However, ablation of DNMT1 in OL blocks the growth of OL progenitors [36]. Moreover, DNMT 1 deletion is known to be lethal in mammals [37] since proliferating cells require a stable epigenetic environment maintained by DNMT 1. The absence of DNMT 1 triggers cell apoptosis [38,39]. DNMT 3A ablation in transgenic mice induced OL differentiation defects and a reduced functionality to remyelinate axons following injury [40]. DNA methylation and hydroxymethylation were detected at a higher level in adult OLs compared to adult OPCs. Amplified hydroxymethylation is required for spinal cord myelin repair in young mice [32]. On the contrary, senescence-dependent mitigation of hydroxymethylation resulted in irregular gene expression causing ineffective myelin repair, proven by the incidents of abnormal swellings at the axon–myelin interface.

In neuronal progenitor cells, the DNA demethylases TETs are necessary for differentiation, axonal growth, and functional neuronal network formation [41]. TET expression in glial cells has been well established, but a lot remains to be unraveled as to what the underlying epigenetic status is. TET1 is a major enzyme propelling DNA hydroxymethylation in oligodendroglia cells in the spinal cord and its level declines with aging [32]. Other enzymes, including TET2 and TET3, though expressed, do not play a significant role. A major reason as to how and why TET1 controls this major phenomenon in the optic nerve could be the presence of a “CXXC” domain, enabling the protein to bind directly with DNA and that is absent in other TETs [42]. An alternative cause could be the distinct protein binding partners of TETs that triggers enzymatic activity. TET1 and TET2 mutant models exhibit faulty developmental myelination, but it is imperative for adult OPCs to activate hydroxymethylation catalyzed by TET1 and subsequent downstream gene expression for a successful differentiation process. In TET2 and TET3 deficient (KO) mice, early-born retinal cell types—RGCs and amacrine cells (AC)—were affected in development [43]. RGCs in TET2 and TET3 KO mice expressed *Zn8*, a marker of cell terminal differentiation but were restricted to the central retina, and the optic nerve in these zebrafish groups was deformed and even absent in a subset [43]. Reduced numbers of Amacrine cells and differentiated red/green double cones were also noted in TET2 and TET3 KO mice and localized only in the central retina. Together, the presence of the DNA demethylation mechanism is a vital phenomenon in normal retinal development and this knowledge can be useful for understanding the altered state of epigenetics during adulthood and disease.

Epigenetic changes are not only limited to the nucleus but can also mediate the functioning of other cellular organelles, such as proteins in the cytoplasm [44] and mitochondria [45]. This in turn affects common cellular events such as apoptosis [46], autophagy [47], inflammation [48] etc. However, to date, very limited groups have studied the relationship between epigenetic modifications, mitochondrial function, and cytoplasmic epigenetic modification in relation to optic nerve development, regeneration, and repair. It is a topic of research whose therapeutic potential remains untapped. A recent comprehensive review discusses in detail the role, interdependency, and localization of epigenetics events in eukaryotic cells [49]. One key study shows lutein endorses neural differentiation, possibly in a PI3K-AKT-dependent manner accompanied by enhanced glycolysis and mitochondrial function [50]. This fuels the synthesis of Acetyl-CoA (an essential ingredient for epigenetic modulation) consistent with epigenetic-based changes in the transcriptome that facilitates neuronal differentiation. Another study understanding myopia discusses plausible cytoplasmic proteins and pathways (Wnt signaling, protein kinase/growth factor signaling, and IGF-1 signaling) regulated by DNA methylation in relation to various ocular cells/tissues [44].

Apart from the addition and removal of a methyl group to the nuclear content, the addition and removal of the acetyl group in histones play a vital role in reconfiguring the chromatin structure and resulting gene expression profile. As described in the previous section (Section 2), HDACs are a major regulator of gene expression, and they act by removing the acetyl group from the histone structure and the family of HDACs has been extensively studied for optic nerve repair and regeneration.

HDACs are known to regulate glial cell development and pathologies of the CNS; however, research studying the developmental expression and functionalities of HDACs in the developing optic nerve and retina are limited. HDACs 1, 2, 3, 5, 6, 8, and 11 are all locally situated to the nuclei of glia during the development and maturation of the optic nerve [21]. HDACs 1 and 2 localize primarily in the nucleus, HDACs 3, 5, 6, 8, and 11 are detected mainly in the cytoplasm and nuclear region during more than one stage of development, and HDACs 4, 9, and 10 are cytoplasmic in all stages of development [21]. These data are the critical initial step in identifying HDAC-associated functions that may plausibly modulate chromatin reconfirmation during differentiation and regeneration of the optic nerve in development and disease processes and pave the way in understanding optic nerve pathologies where localization of HDACs is integral in disease progression.

Other glial cells in the optic nerve are immensely critical for the maintenance of neuronal functioning in the CNS, including astrocytes and microglia. Not much has been conducted directly linking the modulation of these cells and their epigenetic changes in optic nerve pathologies. However, earlier reports using alternative models have established the role of epigenetic factors such as HDACs in these cells. In response to inflammatory conditions, HDAC activity is heightened in the astrocytes. Conjointly, glial inflammatory responses in microglia and astrocytes are mitigated following HDAC activity inhibition [51,52]. In primary human astrocytes, glial fibrillary acidic protein (GFAP) is upregulated in reactive astrocytosis but reduced after HDAC inhibition [53] without affecting astrocyte activation [54]. HDAC inhibition in various CNS injuries reduces the upregulation of IL-1β, cyclooxygenase (COX)-2, iNOS, and TNF-α in reactive astrocytes [54]. Production of glycosaminoglycans like chondroitin sulfate proteoglycan (CSPG) can be increased by reactive astrocytes, thereby lowering acetylation levels in neighboring cells, as they can act as HAT inhibitors [55]. A study by Kuboyama showed that HDAC3 was highly upregulated in a contusion spinal cord injury (SCI) model in microglia/macrophages [56]. Up to date, there has been no report directly examining the behavior of glial cells in the optic nerve following epigenetic alterations, and this topic remains to be explored comprehensively.

Rao et al. showed that histone lysine methylation (e.g., H3K9me2 and H3K27me3) and associated expression of the respective histone methyl transferases, G9a, and enhancer of zeste homolog 2 (Ezh2), occurs in RGCs during retinal development [22]. Moreover, the study also showed inhibition of Ezh2 or G9a is associated with RGC death, thereby cementing the importance of histone methylation patterns in parts of the optic nerve. A recent study also showed that upregulation of Ezh2 is necessary for spontaneous axon regeneration of sensory neurons in different models [25]. Ezh2 does so by downregulating synaptic function-related genes, including *Slc6a13*, which encodes GABA transporter 2. Expression patterns of basic helix-loop-helix (bHLH) transcription factors such as atonal homolog 5 (*ATH5*) and *NeuroM*, *NeuroD*, and *β3* genes, rely heavily upon histone methylation as methylation of histone H3 at *NeuroM* and *NeuroD* promoters regulates RGC development [57].

## 4. DNA Modification in Optic Nerve Repair and Protection

Several recent studies are trying to shed light on integral epigenetic mechanisms during optic nerve damage and diseases. These findings hold the key to understanding the early events that trigger disease progression and limit the optic nerve’s abilities to regenerate following trauma or disease. Some of the most prominent models that are used to comprehend optic nerve damage and related epigenetic modifications include optic nerve crush (ONC) [12], microbead model induced glaucoma [12], streptozotocin (STZ) induced diabetic insult [58], etc. DNA methylation and histone acetylation levels have been abundantly investigated in these studies. The data obtained can help researchers design targeted and efficient therapeutic tools and a recipe for tailoring the epigenetic system in the tissue for successful regeneration and protection.

In lower vertebrates, CNS neurons, including RGCs, regenerate axons throughout their lifetime; however, other neuronal types, such as hindbrain neurons in tadpoles, lose the capacity to regenerate post spinal cord injury [59]. This model offers a unique opportunity to explore genes involved that are responsible for regenerative or non-regenerative responses after CNS injury [59]. Whole genome bisulfite sequencing (WGBS) from animals during optimal axon regeneration time point demonstrated that DNA of regenerative CNS is more accessible. Reduced DNA methylation status was observed in regenerating tadpole hindbrain and frog eye relative to the non-regenerative state [59]. However, a very paradoxical observation was also made in the study where in regenerative CNS of these models, many genes displayed augmented, promoter-associated CpG-methylation following injury and exhibited increased RNA expression and association histone markers for active promoters and enhancers. This might be due to varying upcoming mechanisms that have been studied recently, such as altered connection with activating or repressive transcription factors and histone modifications and augmented association of genes with the nuclear lamina to facilitate an open chromatin structure [60,61]. In both the CNS and PNS, neuronal gene expression is altered following axonal injury. Transcriptional factors such as neurotrophin Bdnf and Sox11 promote efficient axon regeneration in the PNS, in which nerves regenerate after injury, but not in the CNS of mammals [62]. Both these factors are highly influenced by DNA methylation.

Streptozotocin (STZ) is a known inducer of diabetes-related pathologies, including diabetic retinopathy. Multiple administration of STZ increased global DNA methylation in the retina, resulting in RGC damage [63]. Six weeks post-STZ injection, the levels of DNMT 1 and DNMT 3B increased notably relative to control vehicle-injected mice. In RGCs of diabetic retinopathy, DNMT 1 was found to be intensely upregulated, and DNMT 1 modulated DNA methylation was also associated with diabetic retinopathy progression. Mice subjected to ONC displayed an overall reduction in histone acetylation in the RGC layer as early as 24 h post-crush, which is reflective of the decreased expression of several associated genes.

The study mentioned earlier studied the roles of *Oct4, Sox2, and Klf4* (OSK) co-expression in reversing the aging process demonstrated in a glaucomatous animal model of increased axonal density relative to control mice that received no OSK [12]. It suggests a regenerative event mediated after DNA demethylation. OSK-mediated increase in Stat3 mRNA levels in promoting RGC survival and axon regeneration depends on the activities of TET1 and TET2. However, DNA demethylation is not the only factor required for RGC protection and axon regeneration, as overexpressing the TET1 catalytic domain by itself was not successful in promoting axon regeneration. These data indicate a complex intertwined and time-dependent epigenetic machinery that controls DNA methylation during optic nerve growth and homeostasis.

## 5. Histone Modifications in Optic Nerve Repair and Protection

The histone acetylase (HAT) and HDAC interdependency appear to tilt the balance toward deacetylation in retinal degenerative diseases [64], and hence the effects of HDAC inhibitor (HDACi) treatment have been studied comprehensively in recent years for restoring equilibrium. HDACi is shown to have a neuroprotective effect when treating damaged retinas or differentiated neurons (Figure 2). HDACs are also known to mediate the deacetylation of non-histone proteins, including microtubules, transcription factors, and even enzymes. For instance, axonal injury causes tubulin deacetylation that is mediated by HDAC5 in DRGs and serves as a prerequisite for regenerative growth [65]. Additionally, blocking HDAC5 promoted the acetylation of microtubules and enhanced DRG growth. The data suggest that finely balanced DNA acetylation of cytoskeletal and structural protein-related genes is critical for successful axonal regeneration. Pan inhibition of HDACs can prove to be deleterious and, therefore, selective inhibition of HDACs can prove useful, as shown in a study where targeted inhibition of HDAC6 ameliorated CNS injury characterized by oxidative stress-induced neurodegeneration and insufficient axonal regeneration [66].

Following ONC, HDAC3 translocated to the nuclei in injured RGCs as a consequence of axonal injury and caused extensive H4 deacetylation and transcriptional dysregulation that resulted in RGC death. In another study, following an acute optic nerve injury, the mRNA levels of class I HDACs, such as HDAC2 and HDAC3, were upregulated and peaked at 72 h in RGCs post-ONC. HDAC3 translocated from the cytoplasm to the nuclei by day 5; an observation consistent with the earlier research where HDAC3 localized to the nuclei in dying cortical neurons in an in vivo Huntington’s disease model. Conditional knockout of HDAC3 or pharmacological administration of RGFP966 blocked HDAC3 activity and improved RGC survival in a dose-dependent manner by preventing nuclear atrophy and apoptosis [24,46]. Mere targeted removal of HDAC3 was not potent enough to provide RGCs protection from axonal degeneration or somatic cell death in a glaucoma mouse model. Studies in aged or chronic glaucoma mouse models further demonstrated that using RGFP966 to inhibit HDAC3 activity provided limited protection against somatic cell loss in the ganglion cell layer. A single intravitreal injection of RGFP966 followed by selective blocking of HDAC3 ceased histone deacetylation, heterochromatin formation, apoptosis, and DNA damage post-ONC [46]. Repeated IP administration of RFGP966 prevented RGC loss, proving the importance of DNA acetylation in retinal pathology.

On the other hand, a detailed investigation of epigenetic regulation of oligodendrocytes in the optic nerve remains vague. In an adult zebrafish optic nerve transection model, the process of olig2 positive cells stays undamaged and the total number of olig2 + cells in NFL is not significantly altered [69]. Moreover, HDAC inhibitors (MS-275, M334, and suberoylanilide hydroxamic acid) are known to improve neuronal differentiation and inhibit oligodendrocytes production [70]. In a few reports, they are shown to cause cytotoxic in oligodendrocyte precursor cells [71]. Therefore, the use of these compounds must be carefully regulated as they may present a double-edged sword.

Apart from direct inhibition using synthetic HDAC inhibitors, there are other factors that modulate HDAC activities. Therapeutic effects of mood stabilizers, lithium [72] and valproic acid (VPA), were reported in retinal and optic nerve injury models [73,74]. VPA functions by directly inhibiting the activity of HDAC and causing histone hyperacetylation. Some of the earlier reports revealed that abnormal histone acetylation/deacetylation might relate to RGC damage in glaucoma. Trichostatin A (TSA), a broad-spectrum HDAC inhibitor, promotes neurite outgrowth and neuroprotection along with neuronal differentiation and neurite branching [64]. The same effect was reported in RGCs through histone H3K9 acetylation. SNC-121, a selective ligand that activates the δ-opioid receptor, has shown RGC neuroprotective effect in glaucoma mice model by regulating the expression and activity of HDACs, increasing acetylation of histone (H3, H4, and H2B), and reducing the activity of class I and class IIb HDAC [75]. HDAC 1 & 2, and SIRT1 (a member of the Class III family of HDACs), are plausible p53 deacetylases [76,77,78]. This entire interplay is acetylation-site and cell-type specific. Double knock out of HDAC 1 & 2 in RGCs in an optic nerve transfection model exhibited a neuroprotective effect [68]. *PUMA*, a novel proapoptotic gene induced by p53, is strongly activated in axotomized RGCs and is also inhibited following HDACI/II ablation, making HDACI&II specific targets for designing the blueprint of neuroprotective therapies [68]. “CREB Binding Protein” (CREBBP, CBP or KAT3A) and “Adenovirus E1A-associated 300-kD Protein” (p300 or KAT3B) are both KAT3 family members that are well recognized for catalyzing acetylation of all core histones. P300 in RGCs are developmentally controlled and their expression remains downregulated post optic nerve injury [23]. It was reported that the regeneration programming after an optic nerve injury relied on the expression of p300, which upregulates acetylation of both histone and non-histone target genes. The Bromodomain and Extra-Terminal Domain (BET) family of proteins is identified by the presence of two tandem bromodomains and an extra-terminal domain. BET family of proteins are encoded by paralogous genes and are made of BRD2, BRD3, BRD4, and BRDT. Bromodomains have the potential to precisely bind acetylated lysine residues in histones and serve as chromatin-targeting modules that decode the histone acetylation code [79]. Hence BET proteins have a pivotal role in modulating gene transcription by altering interactions between bromodomains and acetylated histones during different cell stages of proliferation and differentiation. JQ1, a highly specific blocker of BET proteins, was tested on an acutely damaged RGC model induced by NMDA excitotoxicity [48]. Intravitreal JQ1 administration maintained RGC number, gene expression (including inflammatory genes- *MCP-1, TNFα, RANTES, IL-1β*), and decreased NMDA-induced TUNEL-positive cells in the RGC layer in an animal model.

Another aspect related to post-translational modifications of histone is histone methylation, which mainly occurs on the side chains of lysine. Gene transcription heavily relies on the methylation of histone polypeptides contingent on whether it is mono- di- and tri-methylated. The dimethylation of histone 3 at lysine 9 (H3K9Me2) has been recognized as a chromatin silencer, and it is specifically catalyzed by G9a, a histone methyltransferase [80]. Increased H3K9Me2 has been demonstrated to limit the binding of transcription factors to the promoters of their downstream genes and thus diminishes their further expression [81]. Moreover, G9a has also been found to be significantly expressed in adult mouse retinas and throughout the development. In a traumatic brain injury (TBI) model, increased expression of G9a and H3K9Me2 were noted in RGCs and optic nerves which underwent cell death and oxidative stress [80]. Administration of G9a inhibitor (UNC0638) attenuated H3K9Me2 activity in both optic nerve and RGC and subsequently activated Nrf2 to block oxidative stress. This leaves no doubt that epigenetic regulation plays a pivotal role in retrograde transportation of axons and providing neuroprotection post TBI. The histone methyltransferase catalyzes the tri-methylation of histone H3 at lysine 27 (H3K27me3) to establish a repressive chromatin structure, enhancer of zeste homolog 2 (Ezh2), which is transitorily expressed in the perinatal retina, especially in the RGCs [22,82]. Though Ezh2 does not mediate retinal ganglion cell homeostasis or their susceptibility to injury [82], progressive photoreceptor degeneration was found to be associated with the deletion of Ezh2 from retinal progenitors at the embryonic stage [83]. Cell death in RGC and NMDA-induced inner nuclear layer (INL) was significantly prevented by 3-deazaneplanocin A (DZNep), an inhibitor of transcription of Ezh2 [84]. Moreover, it conserved RGC functionality as shown by maintaining the ERG b/a wave ratio and the b and a-wave amplitudes in NMDA-treated mice. H3K27me3 affects the survival of RGCs at specific transcriptional and epigenetic levels. The absence of H3K27me3 was found to be neuroprotective, as demonstrated by the upregulation of neuroprotective genes in RGCs. Therefore, DZNep, which inhibits Ezh2 activity, could hold the key to novel therapeutic treatment for ocular neurodegenerative diseases. One study looked at lysine-specific demethylase 1 (LSD1-transcription repressor) and its role in the removal of a methyl group from methylated lysine 4 of histone H3 [85]. Tranylcypromine, a major LSD1 inhibitor, repressed neuron cell death post glutamate neurotoxicity and oxidative stress exposure in an NMDA-induced toxicity model. Tranylcypromine overturned the significant glutamate suppression of p38 MAPKc, presenting neuroprotection. Intravitreal administration of tranylcypromine rescued a significant number of RGC in the same model, indicating epigenetic regulation dictating the survival of RGC via the up-regulation of p38 MAPKc activity.

The interaction of transcriptional factor and chromatin accessibility controls the expression levels of several downstream molecular players, such as Gap43 [86] and Tubb3 [87], which localize and function effectively at the growth cone [13]. Most of these downstream genes mediate signaling pathways that control cell metabolism. Several influential pathways that propel cell growth and axon regeneration include deletion of *PTEN*, *IL22*, or *SOCS3* to activate mTOR and *STAT3* pathways [88,89,90]. Additionally, JAK/STAT pathway, an established molecular event in optic nerve hemostasis and regeneration, is also known to be influenced by epigenetic modifications [91]. However, a lot of research still needs to be done to establish a concrete relationship between these cellular mechanisms in the optic nerve.

## 6. Conclusions

Epigenetic modifications are critical for all biological mechanisms driving development, homeostasis, and repair of the optic nerve (Table 1). Designing novel therapeutics which can modify the epigenetic setting for modulating the expression and transcriptional activities of vital genes in the optic nerve is the need of the hour. Since epigenetic modifications occur far upstream during the pathological molecular incidents, any adjustment to its characteristics holds the potential to define and alter the entire downstream and ultimate consequence of the pathological event. Therefore, discovering and understanding its machinery in optic nerve regulation must be carried out with high accuracy before coming to any conclusions and pitching novel epigenetic inhibitors as drug options. For instance, HDAC inhibitors seem to be an optimal treatment for RGC rescuing; however, one must realize multiple substrates of HDACs are involved in various biological events, including differentiation, proliferation, and apoptosis; their expression varies at different stages of development and disease progression. In line with these recommendations, earlier reports exploring HDAC inhibitors have also reported side effects like thrombocytopenia, neutropenia, anemia, fatigue, and diarrhea [92,93]. Therefore, the use of HDAC inhibitors to rescue and regenerate optic nerve must be carried out with optimal precision. Moreover, over 14 different HDAC inhibitors are involved in clinical trials for cancer treatment, but there is a risk of cells gaining resistance to these drugs, and these are drawbacks that might translate to optic nerve research as well. A lot of studies and research still need to be conducted as newer epigenetic factors and players are still being discovered in recent times. Epigenetics may hold the key to successful therapeutic options to address optic nerve-associated diseases.

## Figures and Tables

**Figure 1 ijms-23-08927-f001:**
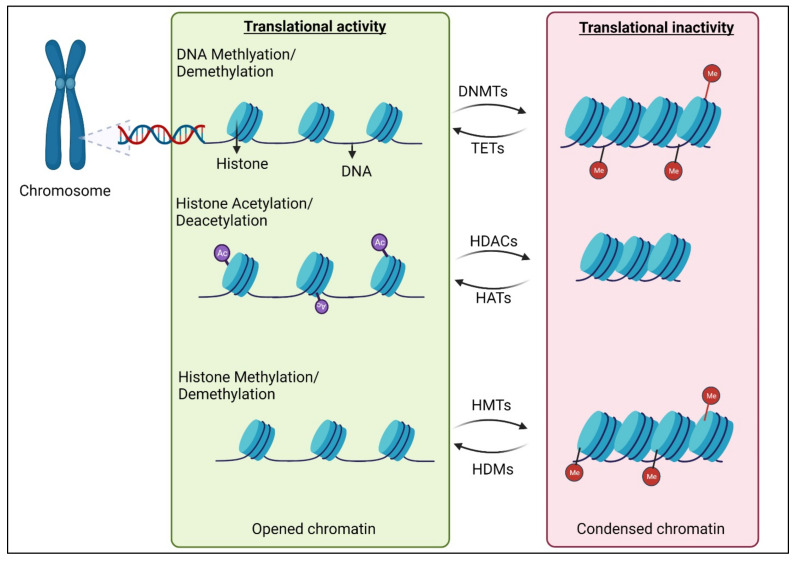
**Epigenetic regulation of gene expression.** Enhanced translational activity occurs after chromatin structure attains an open configuration, generally following DNA and histone demethylation and histone acetylation. This dynamic epigenetic mechanism is fueled by a group of enzymes that are categorized by their characteristic of presenting or reverting the methyl and acetyl groups on the histone or DNA structure. DNMT: DNA methyltransferase; TET: Ten-eleven translocation enzymes; HDAC: histone deacetylase; HAT: histone acetylase; HMT: histone methyl transferase; HDM: histone demethylase; Me: methyl group; Ac: acetyl group.

**Figure 2 ijms-23-08927-f002:**
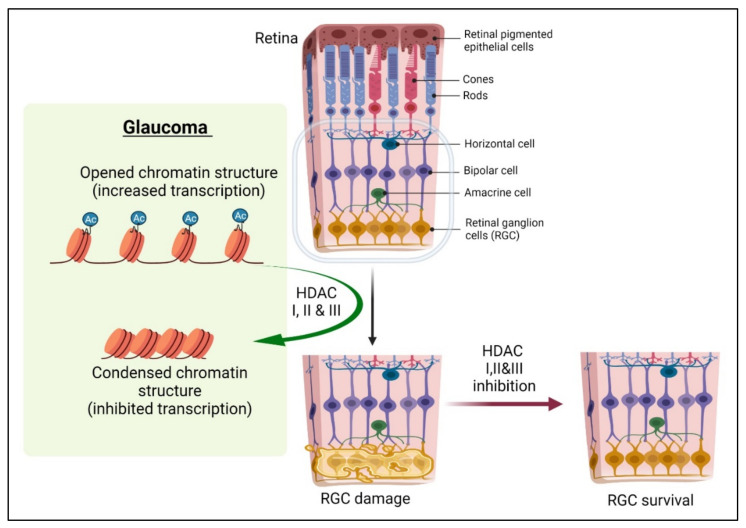
**HDAC inhibitors rescue RGCs by modulating the histone acetylation levels.** HDAC families I, II, and III are known to upregulate significantly in glaucoma models [64]. Blocking HDAC activity using inhibitors offers significant neuroprotection, including enhanced RGC protection. HDAC activity inhibition reprograms the chromatin structure to a pro-translational configuration which occurs through modulation of factors such as p53, CREB-binding protein/p300 (CBP/p300), and the p300-CBP-associated factor (P/CAF), which facilitate neuroprotection, etc. [67,68]. Ac: acetyl group.

**Table 1 ijms-23-08927-t001:** Epigenetic Studies in Optic Nerve Tissues.

Cell Type	Epigenetic Player	Experimental Model	Effect	References
**RGCs** **(Retinal** **ganglion cells)**	TET1-dependent deletion of PTEN	Optic nerve crush model	Optic nerve regeneration	[94]
	Class I HDACs and HDAC 2 & 3 upregulation and HDAC3 nuclear localization in RGCs	Optic nerve crush model	Optic nerve degeneration/RGC apoptosis	[64]
	Increased G9a expression and H3K9Me2 activity in the retina (RGC) and optic nerve	Traumatic brain injury (TBI) model	TBI causes apoptosis and oxidative stress in the retina (RGC) and optic nerve	[80]
	Inhibition of retinal HDAC activity (post valproic acid treatment)	- RGCs purified from new-born (postnatal day P0–P2) rat retinas - HeLa cells	Neuroprotection and histone hyperacetylation	[73,74]
	Inhibition of HDAC3 activity (RGFP966 activity)	Optic nerve crush model	RGC survival and repression of the apoptotic gene in RGCs post optic nerve injury	[24,64]
	Double knock out of HDAC1&2	Optic nerve axotomy	Anti-apoptosis and neuroprotection effect	[68]
	3-deazaneplanocin (DZNep) inhibits Ezh2 inhibition using 3-deazaneplanocin (DZNep) -reduces the trimethylation of histone 3 lysine 27 (H3K27me3) or activity	Retinal/RGC damage caused by intravitreal injection of N-methyl-D-aspartate (NMDA)	Prevent cell death and inner nuclear layer thinning induced by NMDA and improved visual function	[84]
	Increased Histone H3K9 acetylation using Trichostatin A (TSA)	Lead-induced neurotoxicity	Promotes neurite outgrowth and branching, neuroprotection, neuronal differentiation, and neurite branching	[95]
	Intravitreal JQ1 (BET inhibitor) administration	RGC damage induced by NMDA excitotoxicity	Sustained RGC number and gene expression and decreased TUNEL-positive cells in the ganglion cell layer	[48]
	Promotion of p38 MAPKc activity and intravitreal administration of tranylcypromine (lysine-specific demethylase 1 (LSD1) inhibitor)	NMDA-induced excitotoxicity	Enhanced RGC survival	[85]
	OSK-mediated vision restoration is TET1/2 dependent ectopic expression of Oct4 (also known as Pou5f1), Sox2, and Klf4 genes (OSK) in RGC	Optic nerve crush model	Axon regeneration	[12]
**OPCs** **(Oligodendrocyte progenitor cells)**	Increased levels of H3K27me3 from NSCs (neural stem cells) to immature OL and significantly decreased levels of histone acetylation (i.e., H3K9ac) at the early stages of OPC differentiation associated with increasing levels of H3K9me3 during OPC maturation	Human pluripotent stem cell culture	Differentiation of OPCs into OLs	[96]
	DNMT1 downregulation in oligodendrocytes	- Conditional mutation in the mouse DNMT1 gene in embryonic stem (ES) cells. - T24, a human bladder transitional carcinoma-derived cell line; A549, a human non-small-cell lung carcinoma cell line, and NIH 3T3, a mouse fibroblast cell line	- Inhibition of OPC growth - Cell apoptosis - Mild impact on myelin reparation process	[38,39]
**Myelin**	Downregulation of TET1	- TET1 KO mice model	- inefficient myelin repair and axo-myelinic swellings	[32]
			- alters astrocyte morphology and impairs neuronal function	[97]

Disclaimer: All illustrations were created using Biorender (biorender.com).

## Data Availability

Not applicable.

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
