# Peer review of "Epigenetic Regulation of Optic Nerve Development, Protection, and Repair"

_ijms, 2022, doi:10.3390/ijms23168927_

Round 1
Reviewer 1 Report
This is an excellent review that contains much information that will be of interest to a wide range of retinal researchers. There are a few points where I feel the authors could add material to make the article even more useful.
1) The authors should consider creating a separate section for the discussion on glia. Also, the discussion on oligodendrocytes focuses on development. What happens to mature oligos when they are treated with the agents described elsewhere in the text.
2) The authors should expand the discussion of cytoplasmic effects of some of the HDACs and other enzymes. How much of the effects of drugs is due to actions in the cytoplasm.
3) What do the epigenetic modifiers actually do? The authors mention a few specific effects but there are a number of RNAseq surveys of changes induced by some of these agents and some discussion of this would be very useful.
4) Since the authors mention therapeutic agents, it would be helpful to the reader to be told about the unwanted side effects of some of these agents, since this may limit there therapeutic usefulness.
Author Response
We would like to thank the reviewer for their useful comments. We have addressed all comments to the best of our abilities and we hope it satisfies the reviewer concerns.
Reviewer 1
- The authors should consider creating a separate section for the discussion on glia. Also, the discussion on oligodendrocytes focuses on development. What happens to mature oligos when they are treated with the agents described elsewhere in the text.
Response: We appreciate the reviewer’s comment and have added the paragraphs on optic nerve glial cells (Lines 222-237 and 338-344)
- The authors should expand the discussion of cytoplasmic effects of some of the HDACs and other enzymes. How much of the effects of drugs is due to actions in the cytoplasm?
Response: We have addressed the comments in the manuscript (Lines 188-202)
3) What do the epigenetic modifiers actually do? The authors mention a few specific effects but there are a number of RNAseq surveys of changes induced by some of these agents and some discussion of this would be very useful.
Response: We have addressed the comments in the manuscript (line: 413-421)
4) Since the authors mention therapeutic agents, it would be helpful to the reader to be told about the unwanted side effects of some of these agents, since this may limit their therapeutic usefulness.
Response: We have addressed the comment in the manuscript and highlighted the same- line: 340-343 and 434-436.
Reviewer 2 Report
The authors of the Review article "Epigenetic regulation of optic nerve development, protection, and repair" present a revision of the epigenetic regulators that modulate the optic nerve development and regeneration. This would help discovering novel effective drugs switching-on or off epigenetic programs to heal eye illnesses involving the optic nerve (e.g., glaucoma, atrophy, etc). The Review is very well organized and written. It starts with an introduction to the term ‘epigenetics’ and its importance in tissue protection and repair, in particular, the optic nerve, a vital tissue for visual capabilities. The following section describes the molecular mechanism of DNA methylation and histone acetylation/methylation and how it affects chromatin structure. Then, the authors relate these epigenetic modifications to optic nerve ageing and disease, adding references to several studies. In the next section, they review how DNA methylation affects optic nerve repair and protection, and they dedicate another section to histone modifications in optic nerve regeneration. In the conclusion, they mention that some epigenetic drugs are being tested in clinical trials for cancer treatment, however much remains to be done in this regard to develop effective epigenetic treatments for optic nerve related diseases. The Review includes two explanatory figures, the first one showing how epigenetic factors regulate gene expression, and the second one shows the role of epigenetic inhibitors in glaucoma. Finally, there is also one table summarizing the optic nerve epigenetic studies done so far. The Review is very complete, nevertheless, from my point of view; the authors could have added a section to the omics technologies used to study histone modifications (e.g., Chip-seq, CUT&RUN), chromatin accessibility (e.g., DNase-seq, ATAC-seq), chromatin interactions (e.g., Hi-C) and DNA methylation (e.g., EM-seq, Oxford Nanopore long-read sequencing) or if there exists any study using these omics technologies to inspect the epigenome of optic nerve/retina.
Author Response
We would like to thank the reviewer for their useful comments. We have addressed all comments to the best of our abilities and we hope it satisfies the reviewer concerns.
Reviewer 2
- the authors could have added a section to the omics technologies used to study histone modifications (e.g., Chip-seq, CUT&RUN), chromatin accessibility (e.g., DNase-seq, ATAC-seq), chromatin interactions (e.g., Hi-C) and DNA methylation (e.g., EM-seq, Oxford Nanopore long-read sequencing) or if there exists any study using these omics technologies to inspect the epigenome of optic nerve/retina.
Response: We have addressed the comment in the manuscript (line: 66-74)